# Comprehensive Proteomic Analysis Revealed a Large Number of Newly Identified Proteins in the Small Extracellular Vesicles of Milk from Late-Stage Lactating Cows

**DOI:** 10.3390/ani11092506

**Published:** 2021-08-26

**Authors:** Md. Matiur Rahman, Shigeo Takashima, Yuji O. Kamatari, Kaori Shimizu, Ayaka Okada, Yasuo Inoshima

**Affiliations:** 1The United Graduate School of Veterinary Sciences, Gifu University, 1-1 Yanagido, Gifu 501-1193, Japan; matiur.vetmed@gmail.com; 2Laboratory of Food and Environmental Hygiene, Cooperative Department of Veterinary Medicine, Gifu University, 1-1 Yanagido, Gifu 501-1193, Japan; skaori@gifu-u.ac.jp (K.S.); okadaa@gifu-u.ac.jp (A.O.); 3Department of Medicine, Faculty of Veterinary, Animal and Biomedical Sciences, Sylhet Agricultural University, Sylhet 3100, Bangladesh; 4Life Science Research Center, Division of Genomics Research, Gifu University, 1-1 Yanagido, Gifu 501-1193, Japan; staka@gifu-u.ac.jp; 5Life Science Research Center, Division of Instrumental Analysis, Gifu University, 1-1 Yanagido, Gifu 501-1193, Japan; kamatari@gifu-u.ac.jp; 6Education and Research Center for Food Animal Health, Gifu University (GeFAH), 1-1 Yanagido, Gifu 501-1193, Japan; 7Joint Graduate School of Veterinary Sciences, Gifu University, 1-1 Yanagido, Gifu 501-1193, Japan

**Keywords:** cow, late-stage lactation, milk sEVs, nanoLC-MS/MS, protein

## Abstract

**Simple Summary:**

Bovine milk provides excellent nutrition for infants and contains significant bioactive compounds, including small extracellular vesicles. Small extracellular vesicles are nanosized particles that have long been recognized to carry a variety of cargos, including proteins, nucleic acids, lipids, and other biomolecules. Previous research has shown that the proteomic profile in milk small extracellular vesicles differed in accordance with the lactation stages of cows, such as early- and mid-stage lactation. However, the protein profile in milk small extracellular vesicles from late-stage lactation is mostly unknown. In this study, a comprehensive proteomic analysis was performed to explore the proteins in milk small extracellular vesicles from late-stage lactating cows. The present study identified a total of 2225 proteins in milk small extracellular vesicles from late-stage lactating cows. Notably, a total of 429 proteins were newly identified after comparing with the previously published bovine milk small extracellular vesicles proteomic datasets. These findings indicate that the newly identified proteins could be helpful in better understanding the proteomic dynamics in milk small extracellular vesicles during the late-stage lactating cows.

**Abstract:**

Bovine milk contains small extracellular vesicles (sEVs) that provide proteins, miRNAs, mRNAs, DNAs, and lipids to target cells and play a role in intracellular communications. Previous studies have characterized proteins in milk sEVs from early- and mid-stage lactation. However, the proteins in milk sEVs from late-stage lactation are mostly unexplored. The aim of this study was to determine the proteomic profile of milk sEVs from late-stage lactating cows. A comprehensive nanoliquid chromatography–tandem mass spectrometry (nanoLC-MS/MS) approach was carried out to reveal the proteins in milk sEVs. Additionally, bioinformatics analysis was carried out to interpret the molecular signatures of newly identified proteins in milk sEVs from three late-stage lactating cows. NanoLC-MS/MS analysis revealed a total of 2225 proteins in milk sEVs from cows. Notably, after comparing these identified proteins with previously deposited datasets of proteins in bovine milk sEVs, 429 proteins were detected as newly identified. Bioinformatic analysis indicated that these newly identified proteins in milk sEVs were engaged in a diverse range of molecular phenomena relevant to mammary gland physiology, milk production, immunity, and immune response. These findings suggest that the newly identified proteins could expand the inventory application of molecular cargos, nutritional status, and immune modulation of sEVs in milk during the late-stage lactation.

## 1. Introduction

Bovine milk is a widely consumed natural food that provides an excellent source of nutrition for infants in the early stages of development, growth, and health [1]. Extracellular vesicles (EVs) are notable elements of bovine milk and are thought to be essential signalosomes that mediate cell contact between the dam and the infant [2]. EVs are nanoparticles derived from the endolysosomal system; they are released into bio-fluids by almost all cell types and facilitate their regulation of different physiological and pathological activities [3]. There are different classes of EVs, including exosomes, ectosomes, shedding microvesicles, and apoptotic bodies, classified according to their size, biogenesis, and release pathways [3]. An EV isolated by the use of ultracentrifugation and by passing through a 0.22-µm filter belongs to one of the EV subtypes known as “exosomes” [4]. According to the Minimal Information for Studies of Extracellular Vesicles guidelines 2018 (MIEV2018), the use of the term “small EVs” (sEVs) instead of “exosomes” is suggested [3]. The wide availability of bovine milk has hastened the fundamental research on milk sEVs in recent years. Bovine milk sEVs not only provide genetic information to calves and humans but are also implicated in diverse pathophysiological processes, including bioactivity [5], development of the gastrointestinal and immune systems [6,7], drug delivery system [8], and immunotherapy against cancer [9].

The molecular dynamics of milk vary with different factors, including lactation stage, age, breed, nutrition, energy balance, and udder health status [1]. Lactation is a crucial part of mammalian reproductive physiology and involves the secretion of ample quantities of milk by mammary epithelial cells [10]. The lactation cycle is divided into four stages: early-, mid-, and late-stage lactation, where each of the stages lasts roughly 120 days, and a dry period lasting up to 65 days. During lactation, the cow’s mammary gland undergoes many changes, especially in terms of secretory diminution, which is important for the dairy industry, infant nutrition, and health [11]. A significant alteration may occur in the late stages of lactation for the preparation of involution of the mammary gland following the cessation of milk secretion during the drying period and the next new lactation cycle [1,10].

It is worth noting that the alteration of proteomic cargos in milk sEVs from early-, mid-lactation, and commercial milk have been widely investigated [5,8,12,13,14]. However, less attention has been paid to identifying the proteomic profile of milk sEVs from late-stage lactating cows. Moreover, the dynamics of relative abundances of milk sEVs proteins from late-stage lactating cows have never been studied. We hypothesized that the protein compositions in milk sEVs could be different in late-stage lactation compared with early-, mid-stage lactation, or commercial milk. We also predicted that the proteomic profile could be provided novel information about cows’ physiological status that can be linked to late-stage lactation. As a result, detailed identification of proteins with their diverse composition and biological functions in milk sEVs from late-stage lactating cows is urged to understand the lactation process at the molecular level. Hence, the objective of this study was to determine the proteomic profile of milk sEVs from late-stage lactating cows.

## 2. Materials and Methods

### 2.1. Sample Collection

Milk samples were collected from three healthy Holstein–Friesian cows at Yanagido Farm, Gifu University, Japan. The health status of these cows was confirmed by obtaining a history of routine veterinary examinations from the farm’s data book. The clinical data of the three cows indicated that they were in the late-stage lactation, despite their varying ages and parities (Table 1). These three cows were also used in our previous study [15].

### 2.2. Isolation and Characterization of Milk sEVs

#### 2.2.1. Isolation of Milk sEVs

Isolation and purification of milk sEVs were carried out as described previously [15,16] with slight modifications. Briefly, after removing the milk fat by centrifugation at 2000× *g* for 20 min using a centrifuge (MX-307, Tomy Seiko, Tokyo, Japan), defatted milk was preheated at 37 °C for 10 min. For efficient isolation of milk sEVs, acetic acid was added (finally 1%), and casein was removed by centrifugation at 5000× *g* for 20 min. The whey was filtrated by using 1.0, 0.45, and 0.2 μm pore-size filters (GA-100, C045A047A, and C020A047A, Advantec, Tokyo, Japan). Further, milk sEVs were concentrated from the whey by ultracentrifugation (UC) at 100,000× *g* at 4 °C for 1 h using a P42A angle rotor (Hitachi Koki, Tokyo, Japan) in a Himac CP80WX ultracentrifuge (Hitachi Koki). The supernatant was discarded and the bottom pellet was resuspended with phosphate-buffered saline (PBS) up to 10 mL into a 13PET tube (Hitachi Koki). The UC was carried out twice again at 100,000× *g* at 4 °C for 1 h using a P40ST swing rotor (Hitachi Koki) and the milk sEVs pellet was resuspended with 100 µL of PBS for further use.

#### 2.2.2. Characterization of Milk sEVs

According to the Minimal Information for Studies of Extracellular Vesicles guidelines 2018 (MISEV2018) [3], the isolated milk sEVs were characterized biophysically, transmission electron microscopy (TEM), western blotting (WB), and nanoparticle tracking analysis (NTA) as described previously [15] with slight modifications. In brief, the milk sEVs pellet was diluted 10 times from its original concentration using distilled water (DW) and applied to glow-discharged carbon support films on copper grids. The milk sEVs pellet solution was stained by using 2% uranyl acetate and dried. The milk sEVs morphology was visualized under an electron microscope, JEM-2100F (JEOL, Tokyo, Japan) at 200 kV. In WB analysis, the primary antibodies, anti-CD63 (1:400, M-13, SC-31214, Santa Cruz Biotechnology, Santa Cruz, CA, USA) and anti-HSP70 (1:100, N27F3-4, Enzo Life Science, Farmingdale, NY, USA) were used and incubated the membrane at room temperature (RT) for 1 h followed by being washed thrice.

The secondary antibodies, anti-goat IgG donkey antibody (1:2000, SC-3851, Santa Cruz Biotechnology) and anti-mouse IgG sheep antibody (GE Healthcare, Chicago, IL, USA) conjugated with horseradish peroxidase were used and the membrane was incubated at RT for 1 h followed by being washed thrice. Peroxidase activity was detected by using a Pierce ECL Plus substrate (Thermo Fisher Scientific, Waltham, MA, USA), and the image was visualized by using a chemiluminescence apparatus (ChemiDoc XRS+, Bio-Rad Laboratories, Hercules, CA, USA). NTA analysis of milk sEVs was performed by using a NanoSight LM10V-HS, NTA 3.4 instrument (Malvern Panalytical, Malvern, UK) by an entrusted company (Quantum Design Japan, Tokyo, Japan).

### 2.3. Proteomic Analysis

Comprehensive proteomic analysis of milk sEVs from three cows was carried out using nanoliquid chromatography–tandem mass spectrometry (nanoLC-MS/MS) method performed by an entrusted company, Hakarel (Osaka, Japan), as described previously [15]. The detailed procedures are described below.

#### 2.3.1. Milk sEVs Protein Treatment

Milk sEVs pellet solution was washed by adding eight times the volume of cold acetone in it and incubated at −20 °C for 2 h. After centrifugation, the milk sEVs protein was dissolved in a closed ultrasonic disruptor with 100 mM Tris pH 8.5 and 0.5 of sodium dodecanoate (SDoD). The protein concentration was measured by using a bicinchoninic acid protein assay (BCA) (Bio-Rad Laboratories) and adjust the concentration with 100 mM Tris pH 8.5 and 0.5 of SDoD to 1 µg/µL. To cleave the disulfide bond of the milk sEVs protein, the solution was diluted with dithiothreitol to a final concentration of 10 mM and incubated at 50 °C for 30 min. Iodoacetamide (IAA) was added to a final concentration of 30 mM and incubated at RT for 30 min to alkylate cysteine residues.

To stop the IAA reaction, 60 mM cysteine was added to the mixture and incubated at RT for 10 min. To increase enzyme digesting efficiency, 150 µL of 50 mM ammonium bicarbonate was added. For peptide fragmentation of protein, 400 ng lys-C and 400 ng trypsin were added and incubated at 37 °C for overnight. To remove SDoD, acidify the solution with 5% trifluoroacetic acid and then centrifuge at 15,000× g, at RT for 10 min followed by the collection of supernatant. After desalting, the samples were placed in a C18 spin column (Thermo Fisher Scientific) and dried by a centrifugal evaporator. Further, the peptide was dissolved in 3% acetonitrile (ACN) and 0.1% formic acid water in a closed ultrasonic peptide crusher. The peptide concentration was measured by BCA assay (Bio-Rad Laboratories) and adjusted the concentration at 200 ng/µL.

#### 2.3.2. NanoLC-MS/MS Analysis

The peptides were analyzed using an UltiMate 3000 RSLC nano LC system (Thermo Fisher Scientific). An amount of 200 ng of the peptides was loaded into the LC system using a column (75 um × 250 mm), CAPCELL CORE MP (Osaka Soda, Osaka, Japan), core–shell particle (C18, 2.7 um, 16 nm, Thermo Fisher Scientific). Later, peptides were eluted around 30 min using a linear gradient of A solvent containing 0.1% formic acid with DW and B solvent containing 0.1% formic acid with 80% ACN. The Data Independent Acquisition (DIA) mass spectrometry 1 (MS1) method included full scan measurements for components 1 and 3 at 60,000 resolutions, with an AGC target of 3e6 and a maximum injection time (IT) of 150 min. The scan ranges from 495 to 865 m/z. DIA (MS2) measurement parameters for events 2 and 4 were acquired at 15,000 resolutions with AGC target 3e6, and maximum IT was auto. The loop count was 46 for event 2 and 45 for event 4, and the first mass was fixed at 200 m/z. The normalized collision energy was 24%, 26%, and 28%, respectively, and the isolation window was 8.0 m/z. The spectra were captured in profile format.

#### 2.3.3. Scaffold DIA Analysis

The raw data were analyzed using the Scaffold DIA software (Proteome Software, Portland, OR, USA) to compare the peptide counts of the identified proteins (false discovery rate > 1%). To determine the statistical significance of the data, a moderated *t*-test with the Benjamini–Hochberg test [17] was used. Statistical significance was set at *p* < 0.05.

### 2.4. Comparative Data Analysis

Milk sEVs proteins from the present study were compared with previously reported [5,12,14] bovine milk sEVs proteins. The raw data of the milk sEVs proteins were combined to generate a single dataset. Proteins without a gene name or dual-name but same genes were excluded from the dataset. This dataset contains milk sEVs proteins from early-and mid-lactation, and commercial milk from cows. All raw data were loaded into the FunRich software ver. 3.1.4 (Victoria, Australia; http://www.funrich.org/, accessed on 2 December 2020) and analyzed using a Venn diagram [18].

### 2.5. Bioinformatic Analysis

To reveal the potential function and biological insights into the newly identified proteins in milk sEVs from the three cows, Gene Ontology (GO) annotation, including comparative cellular components, biological processes, molecular functions, and biological pathway analyses were carried out using the FunRich software.

## 3. Results

### 3.1. Isolation and Characterization of Milk sEVs

TEM revealed that milk sEVs exhibited a spherical bilayered shape (Figure 1A). WB analysis successfully detected the milk sEV-surface-marker protein CD63 and the internal protein HSP70 (Figure 1B; Appendix A). NTA showed that the peak (mode) intensities for the particle size distribution of the three milk sEVs samples from cows 1, 2, and 3 ranged from 145.7 to 167.1 nm (156.0 nm average mean peak size) (representative sample cow 2, Figure 1C). The particle concentration ranged from 4.23 × 10^10^ particles/mL to 3.82 × 10^11^ particles/mL (data not shown). The results confirmed the presence and enrichment of milk sEVs in the current study.

### 3.2. Proteomic Analysis

Comprehensive proteomic analysis by nanoLC-MS/MS revealed a total of 2225 proteins in milk sEVs from the three cows (Appendix A). The identified proteins in each of the milk sEVs samples from cows 1, 2, and 3 were 1560, 1974, and 1690, respectively (Figure 2). Raw data of proteomic analysis of milk sEVs from cows 1, 2, and 3 were deposited and available in the Mendeley data repository (Mendeley Data, V1, doi: 10.17632/7c2ddgwcgt.1, accessed on 7 October 2020). A total of 1251 proteins were commonly identified in the milk sEVs from the three cows. After comparing the identified proteins among the three cows, it was found that the number of non-overlapping proteins in cows 1, 2, and 3 were 106, 371, and 142, respectively. The current study revealed different numbers of proteins in milk sEVs from the three cows, indicating the variability of milk sEVs biogenesis depending on the individual. 

To identify any known proteins in milk sEVs, the identified proteins were compared with previously published bovine milk sEVs datasets [5,12,14]. After comparing with the bovine milk sEVs dataset, a total of 429 newly identified proteins were discovered in milk sEVs (Figure 3; Appendix A) that were not previously reported in bovine milk. This result indicated that the current study identified a large number of proteins that were apparently new in the dataset of bovine milk sEVs.

### 3.3. Bioinformatic Analysis

GO analysis was performed to reveal the biological phenomena associated with the identified proteins in milk sEVs. Comparative cellular component analysis was carried out to demonstrate the subcellular derivation of the identified proteins in milk sEVs (Figure 4). The milk sEVs from cows 1, 2, and 3 contained a high percentage of proteins derived from the cytoplasm, exosomes (sEVs), and lysosomes, indicating the sEVs enrichment concept. Comparative proteomic analysis revealed 1251 common proteins among cows 1, 2, and 3 (Figure 2). As expected, the milk from all three cows was highly enriched with common milk proteins such as butyrophilin, lactadherin, fatty acid synthase, alpha-lactalbumin, and lactotransferrin. The presence of these proteins indicated the reliability and reproducibility of the current study method for milk sEVs enrichment derived from late-stage lactating cows. 

Finally, the present study investigated the 429 newly identified proteins in milk sEVs that were specific for late-stage lactating cows. To generate a comprehensive image of these newly identified proteins, we examined the enrichment quantity of the 429 newly identified proteins in milk sEVs (top 50) (Table 2). It was found that few biologically important proteins such as serum albumin (ALB), kappa-casein (CSN3), alpha-S2-casein (CSN1S2), major heat shock protein (HSPA1B), Rab protein (RAB27B), immunoglobulin J chain (JCHAIN), and apolipoprotein D (APOD) were highly enriched in the newly identified proteins in milk sEVs from late-stage lactating cows.

The newly identified proteins were involved in a broad range of biological processes such as protein modification (26.25%), xenobiotic metabolism (23.30%), cell proliferation (19.76%), and apoptosis (16.22%) (Figure 5A). As for the molecular function, the majority of newly identified proteins appeared to participate in chaperone activity (25.96%), oxidoreductase activity (5.90%), and protease inhibitor activity (3.83%) (Figure 5B). Biological processes and molecular functions are extensively utilized because they provide a broad data overview, but their capacity to uncover individual signaling pathways is restricted. Therefore, biological pathway analysis of newly identified proteins in milk sEVs was carried out. The biological pathway analysis revealed that the newly identified proteins in milk sEVs mostly participated in metabolism, metabolism of proteins, immune system, and homeostasis (Figure 6). Biological pathway analysis revealed the potential connections between newly identified proteins and the pathways that connect them to biological processes and molecular functions.

## 4. Discussion

sEVs analysis using proteomic technologies is highly useful for better identification of sEVs proteins and molecular signatures from milk. Using large-scale analyses, proteomics can uncover functional data related to the identified proteins, with an emphasis on complex molecular mechanisms and pathways. Therefore, these protein alterations in milk sEVs during late-stage lactation can reflect the physiological and metabolic changes of cows’ health. Thus, the present study aimed to explore the proteomic profile of milk sEVs from late-stage lactating cows, which has not been previously studied.

The present study revealed a total of 2225 proteins in milk sEVs from three late-stage lactating cows, 429 of which were newly identified after comparison with the bovine milk sEVs proteomic dataset [5,12,14]. The current study represents a novel proteomic approach of milk sEVs from late-stage lactating cows, which collects biologically relevant information about the functional role of proteins regarding molecular cargos, nutritional status, and immune modulation. Reinhardt et al. [19] performed the first proteomic analysis of bovine milk sEVs that reported a total of 2107 proteins in milk sEVs from mid-stage lactating cows and provided new information of proteins regarding mammary physiology. In recent years, Samuel et al. [12] and Yang et al. [13] have studied the proteomic analysis of milk sEVs from early- and mid-stage lactating cows and highlighted the biological role of bovine milk sEVs in immune regulation, growth, repair, and development of the infant. On the other hand, proteomic analysis of commercial milk sEVs was recently reported and described the significant role of proteins in milk sEVs for enhance metabolism and bioactive for drug-delivery application. The differences of the diverse variations of milk sEVs proteins between the present study and previous studies due to physio-immunological factors including lactation stage, individual body nutrition, age, somatic cell count in milk, and immunity [12,20]. Another reason for the identification of a large number of new proteins in the current study was the use of raw milk collected from late-stage lactating cows. The current study identified a different number of proteins in each of the milk sEVs from cows 1, 2, and 3. The three cows’ age and parity are quite different, resulting in a variation in the number of proteins in milk sEVs. Since the variability of milk sEVs biogenesis depends on the individual, further study on a universal proteomic profile is required.

The current study identified that milk proteins, including ALB, CSN3, CSN1S2, and APOD were present in high amounts in milk sEVs during the late-stage lactation. These proteins modulate key elements of the immune response, as well as lipid transport. A previous study reported that the quantity of ALB protein was increased during the involution of the mammary gland during the late-stage lactation [21]. CSN3 plays a significant role in the small intestine and influences gut functions, including immune stimulation, mineral and trace element absorption, and host defense against infection [22]. APO protein levels were increased in the milk from late-stage lactating cows, thus contributing to the healthy aging of mammary glands [23]. The abundant presence of these milk proteins suggests that the nutritional qualities of milk and milk products from late-stage lactation have important repercussions for nutrition and immunity.

In the present study, HSPA1B, RAB27B, and JCHAIN were enriched in milk sEVs from late-stage lactating cows. HSPA1B is a chaperone protein that plays a major role in response to cellular stress, cell death, and apoptosis, maintaining the balance between survival and innate immunity [24]. A recent study described that a high enrichment of RAB27B proteins in sEVs suggests that they may have a role in mammary gland development, defense mechanisms, and lactation [25]. The presence of HSPA1B and RAB27B proteins in milk sEVs from the late-stage lactation demonstrated that the mammary gland undergoes major physiological alterations to adapt to the dry period while maintaining immune protection. A study reported that the JCHAIN protein plays a significant role in the modulation of the immune response in infants [13]. The presence of the JCHAIN protein in milk sEVs shows that dietary milk from late-stage lactating cows may provide nutrition that helps to prevent inflammatory disorders and promote health. It has been suggested that during the late-stage lactation, many physiological alterations may occur, such as a decline in milk production and gradual involution of the mammary gland. Thus, the mammary gland may adapt to the condition of maintaining milk production with high nutritional and immunological components.

The GO and biological pathway analyses provided new information on how the mammary gland adapts to late-stage lactation by modifying the functional cargos in the regulation of metabolism, milk yield, enzyme activities, cellular activity and differentiation, inflammatory response, and immunity. Therefore, major biofunctional changes occur in the mammary gland and milk composition involving metabolism, protein modification, apoptosis, and interactions of the immune system. Our current findings are consistent with previous reports that indicated that milk sEVs play an important role in the biosynthesis of milk, lactation physiology, and mammary gland physiology [19,26,27]. Our findings reveal that newly identified proteins in milk sEVs transport cargo are implicated in various biological functions and pathways, as well as several proteins that have not yet been examined or discussed. These newly identified proteins could play roles during the alteration of mammary gland function from the late-stage lactation to the dry period. Additionally, these newly identified proteins are likely to be useful for the development of functional and bioactive assays to modulate the synthesis and transport of milk sEVs, as well as their effects on mammary glands, host physiology, and immune modulation in late-stage lactating cows.

## 5. Conclusions

Proteomic analysis revealed a total of 2225 proteins, 429 of which were described as newly identified proteins after comparison with bovine milk sEVs datasets. The present study helps us to decode the biological and molecular dynamics of newly identified proteins in milk sEVs that are involved in diverse physiological and immunological functions in mammary glands and milk components during the late-stage lactation. Overall, the findings suggested that the identification of such proteins from late-stage lactating cows may be of interest in the emerging field of molecular science, with the potential ability to alter the recipient’s metabolism and immunological response.

## Figures and Tables

**Figure 1 animals-11-02506-f001:**
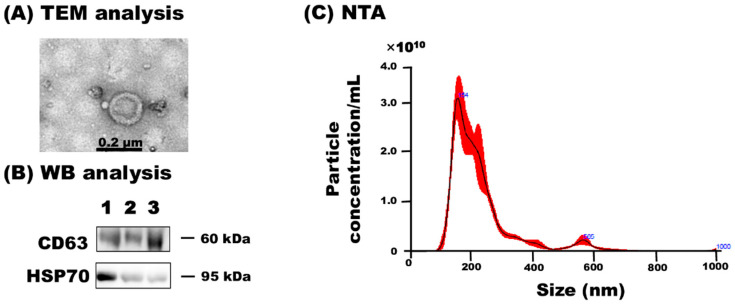
Characterization of milk sEVs. (**A**) TEM detected bilayer spherical shape sEVs in milk (Scale bar shows 0.2 µm). (**B**) WB analysis successfully detected milk sEV-surface-marker protein CD63 and internal protein HSP70 using antibodies, anti-CD63 (SC-31214, Santa Cruz Biotechnology), and anti–HSP70 (N27F3-4, Enzo Life Science), respectively. (**C**) The size distribution of milk sEVs as determined by NTA. A representative data from cow 2 is shown (mean peak size < 0.2 µm in diameter). TEM, transmission electron microscopy; WB, western blot; NTA, nanoparticle tracking analysis.

**Figure 2 animals-11-02506-f002:**
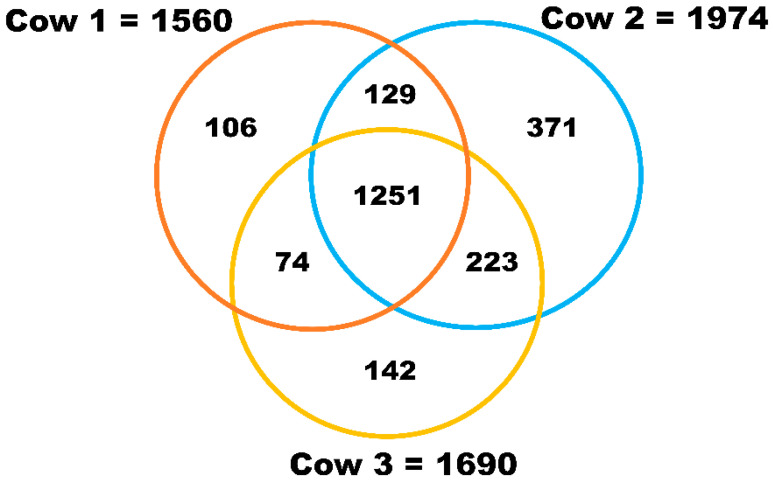
Proteomic analysis of milk sEVs. The Venn diagram displays the number of identified proteins in milk sEVs.

**Figure 3 animals-11-02506-f003:**
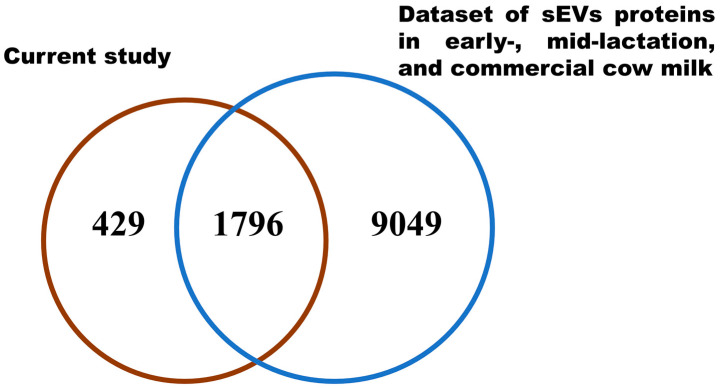
Comparative Vann diagram. Comparison of milk sEVs proteins in the current study with bovine milk sEVs dataset.

**Figure 4 animals-11-02506-f004:**
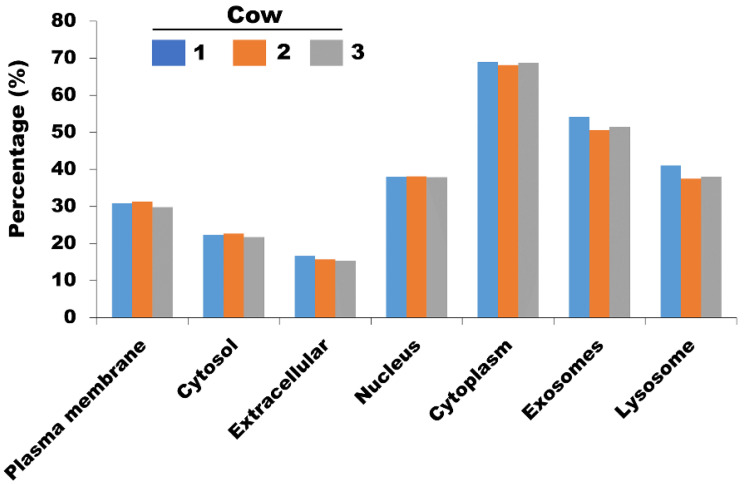
The comparative cellular component of identified proteins in milk sEVs.

**Figure 5 animals-11-02506-f005:**
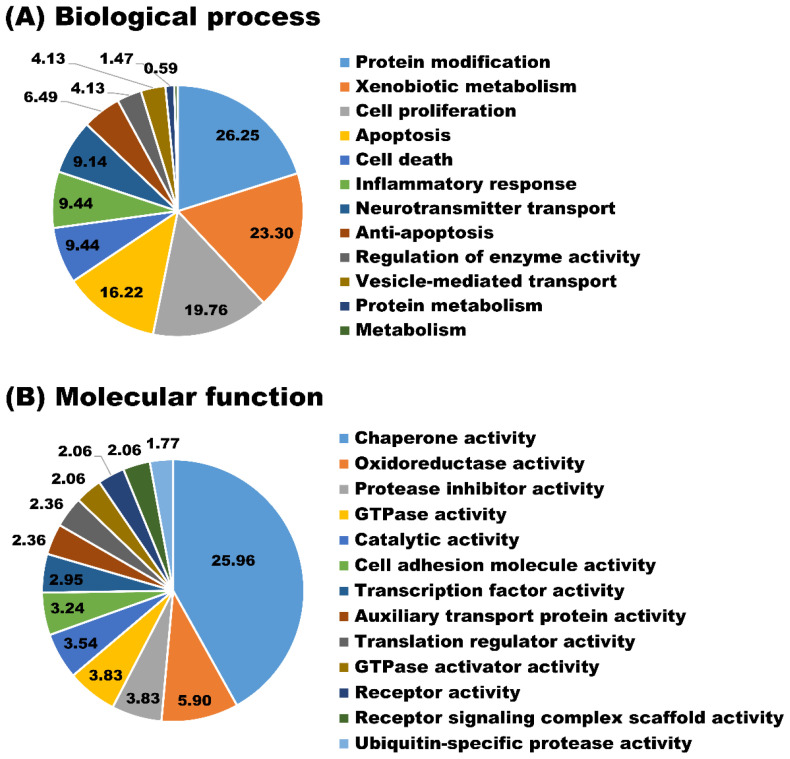
GO analyses. Biological process (**A**) and molecular function (**B**) of newly identified proteins in milk sEVs.

**Figure 6 animals-11-02506-f006:**
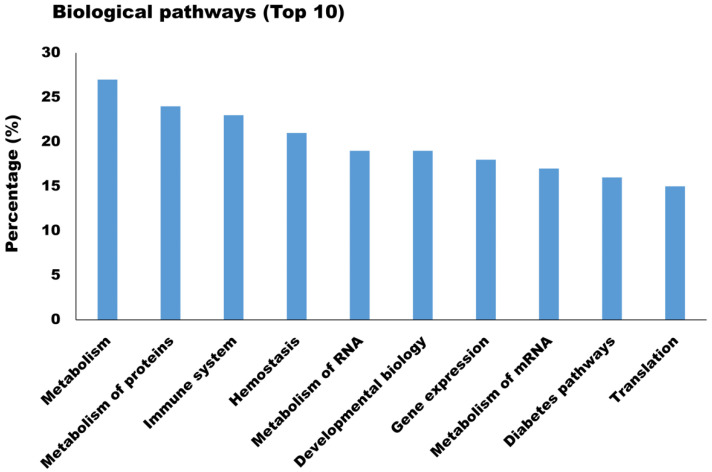
Biological pathway analysis (Top 10) of newly identified proteins in milk sEVs.

**Table 1 animals-11-02506-t001:** Health status of cows used in this study.

Cow	Age (Month)	Number of Parities	Gestation (Week)	Days in Milk	Status
1	111	6	6	310	Healthy
2	89	5	24	256	Healthy
3	54	2	11	302	Healthy

**Table 2 animals-11-02506-t002:** Top 50 newly identified proteins in milk sEVs from late-stage lactating cows.

Protein Name	Gene Name	Quantitative Value			
		Cow 1	Cow 2	Cow 3	Average
Serum albumin	ALB	3.3 × 10^9^	3.7 × 10^9^	1.5 × 10^9^	2.9 × 10^9^
Kappa-casein	CSN3	1.5 × 10^9^	1.3 × 10^9^	7.6 × 10^9^	1.2 × 10^9^
Aldehyde oxidase 3L1	AOX2	4.3 × 10^7^	3.9 × 10^7^	2.7 × 10^8^	9.2 × 10^8^
Alpha-S2-casein	CSN1S2	3.9 × 10^8^	7.0 × 10^8^	6.1 × 10^8^	5.7 × 10^8^
Heat shock 70 kDa protein 1B	HSPA1B	1.6 × 10^8^	1.7 × 10^8^	2.1 × 10^8^	1.8 × 10^8^
Protocadherin gamma subfamily A, 3	PCDHGA3	7.2 × 10^7^	1.6 × 10^8^	2.1 × 10^8^	1.5 × 10^8^
Prostaglandin F synthase	PRXL2B	4.6 × 10^7^	3.1 × 10^7^	9.6 × 10^7^	5.8 × 10^7^
Nucleobindin 2	NUCB2	7.2 × 10^7^	5.8 × 10^7^	3.1 × 10^7^	5.3 × 10^7^
Ras-related protein Rab-27B	RAB27B	4.1 × 10^7^	6.0 × 10^7^	5.3 × 10^7^	5.1 × 10^7^
Immunoglobulin J chain	JCHAIN	4.8 × 10^7^	5.8 × 10^7^	3.7 × 10^7^	4.7 × 10^7^
Collectin-12	COLEC12	4.5 × 10^7^	1.5 × 10^7^	6.1 × 10^7^	4.0 × 10^7^
Pyrroline-5-carboxylate reductase 3	PYCR3	2.7 × 10^7^	4.8 × 10^7^	3.9 × 10^7^	3.8 × 10^7^
Adhesion G protein-coupled receptor F1	ADGRF1	1.4 × 10^7^	3.7 × 10^7^	3.7 × 10^7^	2.9 × 10^7^
NADH-cytochrome b5 reductase 1	CYB5R1	3.0 × 10^7^	1.7 × 10^7^	3.2 × 10^7^	2.7 × 10^7^
Glycerol-3-phosphate acyltransferase 4	GPAT4	8.2 × 10^6^	1.4 × 10^7^	3.7 × 10^7^	2.0 × 10^7^
Selenoprotein F	SELENOF	1.4 × 10^7^	1.4 × 10^7^	2.1 × 10^7^	1.6 × 10^7^
Phospholipid-transporting ATPase	ATP9A	1.2 × 10^7^	7.9 × 10^6^	2.4 × 10^7^	1.5 × 10^7^
Ubiquitin carboxyl-terminal hydrolase MINDY-1	MINDY1	1.6 × 10^7^	1.2 × 10^7^	5.4 × 10^6^	1.1 × 10^7^
ATP-dependent (S)-NAD(P)H-hydrate dehydratase	NAXD	6.2 × 10^6^	1.9 × 10^7^	7.0 × 10^6^	1.1 × 10^7^
Inter-alpha-trypsin inhibitor heavy chain H2	ITIH2	4.0 × 10^6^	2.0 × 10^7^	3.6 × 10^6^	9.2 × 10^6^
Pyridoxal phosphate homeostasis protein	PLPBP	1.0 × 10^7^	5.1 × 10^6^	1.1 × 10^7^	8.6 × 10^6^
Prosaposin	PSAP	4.3 × 10^6^	7.3 × 10^6^	7.6 × 10^6^	6.4 × 10^6^
Apolipoprotein D	APOD	6.7 × 10^5^	1.2 × 10^7^	5.0 × 10^6^	5.9 × 10^6^
Tetraspanin-6	TSPAN6	7.5 × 10^6^	1.5 × 10^6^	2.5 × 10^6^	3.8 × 10^6^
Tubulin-folding cofactor B	TBCB	1.5 × 10^6^	4.6 × 10^6^	4.8 × 10^6^	3.6 × 10^6^
Heme binding protein 2	HEBP2	2.5 × 10^6^	6.1 × 10^5^	2.6 × 10^6^	1.9 × 10^6^
Prefoldin subunit 5	PFDN5	3.3 × 10^5^	9.9 × 10^5^	4.2 × 10^6^	1.8 × 10^6^
RAB3 GTPase activating protein catalytic subunit 1	RAB3GAP1	2.5 × 10^5^	2.3 × 10^6^	2.4 × 10^6^	1.6 × 10^6^
Solute carrier family 7 member 5	SLC7A5	3.5 × 10^5^	3.4 × 10^6^	6.9 × 10^5^	1.5 × 10^6^
Integrin beta-1-binding protein 1	ITGB1BP1	3.3 × 10^5^	2.0 × 10^6^	1.9 × 10^6^	1.4 × 10^6^
Density-regulated protein	DENR	1.1 × 10^6^	2.0 × 10^6^	6.0 × 10^5^	1.2 × 10^6^
Sequestosome 1	SQSTM1	6.5 × 10^5^	1.3 × 10^6^	1.2 × 10^6^	1.1 × 10^6^
Ubiquitin conjugation factor E4 A	UBE4A	7.1 × 10^5^	1.7 × 10^6^	7.8 × 10^5^	1.0 × 10^6^
Phospholipid phosphatase 1	PLPP1	1.5 × 10^5^	3.1 × 10^5^	2.3 × 10^6^	9.4 × 10^5^
Tandem C2 domains, nuclear	TC2N	1.4 × 10^5^	2.1 × 10^6^	5.7 × 10^5^	9.3 × 10^5^
Interferon gamma receptor 2	IFNGR2	1.0 × 10^5^	1.4 × 10^6^	2.9 × 10^5^	9.2 × 10^5^
CD320 antigen	CD320	3.1 × 10^5^	1.4 × 10^6^	8.3 × 10^5^	8.4 × 10^5^
Calcineurin subunit B type 1	PPP3R1	4.4 × 10^5^	9.1 × 10^5^	1.0 × 10^6^	8.0 × 10^5^
Phosphomevalonate kinase	PMVK	1.5 × 10^5^	5.2 × 10^5^	1.3 × 10^5^	7.2 × 10^5^
Glutathione S-transferase Mu 1	GSTM4	4.2 × 10^5^	1.3 × 10^6^	4.1 × 10^5^	7.2 × 10^5^
D-aminoacyl-tRNA deacylase 1	DTD1	9.1 × 10^5^	7.7 × 10^5^	4.3 × 10^5^	7.0 × 10^5^
TRIO and F-actin binding protein	TRIOBP	2.3 × 10^5^	1.2 × 10^6^	6.2 × 10^5^	6.8 × 10^5^
Deoxyhypusine hydroxylase	DOHH	5.4 × 10^5^	7.5 × 10^5^	6.6 × 10^5^	6.5 × 10^5^
Pecanex 1	PCNX1	8.0 × 10^5^	2.1 × 10^5^	3.8 × 10^5^	4.6 × 10^5^
Phosphoinositide-3-kinase adaptor protein 1	PIK3AP1	1.5 × 10^5^	9.7 × 10^5^	2.6 × 10^5^	4.6 × 10^5^
Interferon regulatory factor 3	IRF3	2.5 × 10^5^	4.8 × 10^5^	6.4 × 10^5^	4.6 × 10^5^
Diacylglycerol kinase	DGKD	1.9 × 10^5^	5.3 × 10^5^	3.8 × 10^5^	3.7 × 10^5^
Endoribonuclease LACTB2	LACTB2	2.6 × 10^5^	5.8 × 10^5^	2.6 × 10^5^	3.7 × 10^5^
Ubiquitin like modifier activating enzyme 6	UBA6	2.8 × 10^5^	6.9 × 10^4^	4.9 × 10^5^	2.8 × 10^5^
Nectin-4	NECTIN4	1.8 × 10^5^	2.1 × 10^5^	2.7 × 10^5^	2.2 × 10^5^

## Data Availability

Raw data of proteomic analysis of bovine milk sEVs from cows 1, 2, and 3 were deposited in the Mendeley data repository (Mendeley Data, V1, doi: 10.17632/ 7c2 ddgwcgt.1) (Direct URL: https://data.mendeley.com/datasets/7c2ddgwcgt/1, accessed on 7 October 2020).

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
