# Peer review of "Comprehensive Proteomic Analysis Revealed a Large Number of Newly Identified Proteins in the Small Extracellular Vesicles of Milk from Late-Stage Lactating Cows"

_animals, 2021, doi:10.3390/ani11092506_

Round 1

Reviewer 1 Report

Rahman and colleagues used proteomics technology to analyze small extracellular vesicles in milk cows during late-stage lactation and found 429 new proteins. When done correctly, the newly identified proteins in milk sEVs may help to understand the molecular enigma of late-stage lactation mammary physiology and nutritional dynamics. Unfortunately, the current work from the authors has methodological and scholarly writing issues that require correction.

General comments:

In general, the manuscript lacks an analysis of the research results. The research question is not framed by the introduction. The discussion lacks a critical assessment of the existing literature. Introduction: Differences and associations between whey and SEVS proteomes have not been clearly elucidated. Why not use whey for proteomics instead of SEVS? The significance of studying SEVS is also unclear.

material and method:

Insufficient writing of methods and lack of detailed descriptions of related trials. The purpose of the experiment is not clear.

Result:

There is no need to elaborate too much on the other results. If necessary, please also discuss accordingly.

Discussion:

The discussion of GO analysis section is shallow and fails to explore in depth.

Specific comments:

Page 3: Line 97:Three dairy cows were used as research objects, and the sample size was small, which was not universal.

Page 3: Line 108-116:Lack of sample pretreatment and specific processes for LC-MS/MS

Page 3: Line 100-101:Age and parity have an effect on milk composition and milk yield. Dairy cows with small individual differences should be selected as far as possible. Individual differences should not be ignored and should be explained in the discussion.

Page 3: Line 138-141:The purpose of the NTA trial was unclear and was not discussed in detail. It seems that the NTA did not contribute to the results of the article.

Page 4: Line 158-160:Since the variability of milk sEV biogenesis depends on the individual, the universal significance of this study is challenged.

Page 5: Line 168-169:No subsequent analysis of these proteins is required and no description is required.

Page 6: Line 198-203:The evidence for the selection of these proteins is not well presented nor is it shown in Table 2. The sequence of proteins in Table 2 is suggested to be sorted according to the enrichment abundance.

Page 10: Line 284:These newly discovered proteins should be listed in the supplementary table.

Reviewer 2 Report

Comments to the Corresponding Author:

The manuscript "Comprehensive proteomic analysis revealed a large number of newly identified proteins in milk small extracellular vesicles from late-stage lactating cows” by Md. Matiur Rahman et al analyze the proteins in milk small extracellular vesicles from late-stage lactating 28 cows by nanoLC-MS/MS. The study identified a total of 2,225 proteins, and a total of 429 proteins were newly identified after comparing with the previously published bovine milk small extracellular vesicles proteomic datasets. These findings provided useful knowledge for understanding the proteomic dynamics in milk small extracellular vesicles during the late-stage lactating cows.

The manuscript is a well written manuscript and properly designed scientific work. However, the manuscript requires minor revision. In my opinion, the paper should be published after address some concerns as following:

Del the content from Line 84-94;

Add the days in milk (DIM) for each cow;

Line 141: 4.23×1010,“10” in the latter should in superscript;

Line 142: 3.82×1011, “11” should in superscript;

To improve the quality of Figure 5;

Reviewer 3 Report

This work comes from a respected group that have already contributed significantly in the field of milk extracellular vesicles. In this manuscript by Rahman et al. the authors present a proteomics analysis of isolated bovine milk EVs from late stage lactation. 

As always, this group clearly presents their data and the manuscript is well written. For instance, the amount of identified proteins is very considerable and indicates that the procedures were sound. Additionally, the work provides additional information in the field of milk-derived extracellular vesicles and I therefore support the publication of this manuscript.

I only have a few minor comments:

lines 35-36: provides these molecules to what? The authors should make clear that EVs are a means of intracellular communication, so the listed molecules are provided to target cells.

In the introduction, the authors should describe what the timelines are for early, mid and late lactational stages. 

The authors should provide the entire list of 2,225 proteins as supplementary data (so, an excel sheet that allows easy access and an overview of the identified proteins). In strongly feel that data should be shared upon publication with other researchers! 

The authors describe how the retrieved the data from other studies for their comparison. Also these data should be shared. 

The material and methods section really should be more detailed. First, the isolation procedure should still be shortly discussed (it's easier for readers this way, instead of going to a previous publication). For instance, was it a commercial kit, or density gradient, etc. Also, was the milk pre-processed in order to remove caseins? The same for the proteomics and the EV isolation. I am sure readers would appreciate some more info (I hope the word count allows for this).

I don't get the comparison with the human milk EVs from the Vesiclepedia database, as this study is on bovine milk. As the authors perform their subsequent analysis on the comparison on bovine studies, I strongly suggest that the authors remove the comparison of the human milk proteome (and subsequently correct the text throughout the manuscript). These data are also not needed to support the claims made in this paper, so leaving out this one figure would in my opinion not change the outcome of the study. Moreover, it would make it easier to read (as the authors now basically present two numbers with newly identified proteins, which is slightly confusing).

For the GO-analysis, I would favour some more stringent selection on GO terms that relate to the topic. For instance, in Figure 6 'diabetic pathways' score high. However, I don't see the biological relevance of diabetes in the new born, and I don't believe that there is a strong protective correlation between breastfeeding and diabetes risk. (also, perhaps its better in Figure 6 to have to Go-terms with highest percentage on top, instead of showing these last, in the bottom of the figure). 

Round 2

Reviewer 1 Report

The authors have addressed all my concerns so I recommend the manuscript to be accepted for publication.